


**Ba incorporation in benthic foraminifera**
Lennart J de Nooijer[1]*, Anieke Brombacher[2,a], Antje Mewes[3], Gerald Langer[4], Gernot Nehrke[3],
Jelle Bijma[3], Gert-Jan Reichart[1,2]
[1]Royal Netherlands Institute of Sea Research, Dept of Geology and Chemical Oceanography,
Landsdiep 4, 1797 SZ 't Horntje, The Netherlands
*Corresponding author: ldenooijer@nioz.nl
[2]Utrecht University, Faculty of Geosciences, Budapestlaan 4, 3584 CD Utrecht, The
Netherlands
[3]Alfred-Wegener-Institut Helmholtz-Zentrum für Polar- und Meeresforschung, Biogeosciences
section, Am Handelshafen 12, 27570 Bremerhaven, Germany
[4]The Marine Biological Association of the United Kingdom, The Laboratory, Citadel Hill,
Plymouth, Devon, PL1 2PB, UK
[a]now at: National Oceanography Centre, University of Southampton, Waterfront Campus,
European Way, Southampton SO14 3ZH, UK
**Abstract**
Barium (Ba) incorporated in the calcite of many foraminiferal species is proportional to the
concentration of Ba in seawater. Since the open ocean concentration of Ba closely follows
seawater alkalinity, foraminiferal Ba/Ca can be used to reconstruct the latter. Alternatively,
Ba/Ca from foraminiferal shells can also be used to reconstruct salinity in coastal settings where
seawater Ba concentration corresponds to salinity as rivers contain much more Ba than
seawater. Incorporation of a number of minor and trace elements is known to vary (greatly)
between foraminiferal species and application of element/Ca ratios thus requires the use of





species-specific calibrations. Here we show that calcite Ba/Ca correlates positively and linearly with seawater Ba/Ca in cultured specimens of two species of benthic foraminifera, *Heterostegina depressa* and *Amphistegina lessonii*. The slopes of the regression, however, vary 2-3 fold between these two species (0.33 and 0.78, respectively). This difference in Ba-partitioning resembles the difference in partitioning of other elements (Mg, Sr, B, Li and Na) in these foraminiferal taxa. A general trend across element partitioning for different species is described, which may help developing new applications of trace elements in foraminiferal calcite in reconstructing past seawater chemistry.

Keywords: foraminifera, Ba/Ca, proxies

## 1 Introduction

Incorporation of barium (Ba) in foraminiferal calcite is proportional to seawater barium concentrations (e.g. Lea and Boyle, 1989; 1990; Lea and Spero, 1994). Since open ocean and coastal seawater Ba concentrations correlate well to total alkalinity and salinity, respectively, Ba/Ca of fossil foraminiferal calcite has been used to reconstruct these two parameters (e.g. Lea, 1995; Weldeab et al., 2007; 2014; Bahr et al., 2013). Application of Ba/Ca for such reconstructions, however, critically depends on the prerequisite that temperature and salinity (Lea and Spero, 1994; Hönisch et al., 2011) and photosymbiont activity (Lea and Spero, 1992; Hönisch et al., 2011) do not affect Ba/Ca of planktonic foraminifera. Still, Ba/Ca ratios are known to vary within chamber walls of crust-producing planktonic foraminifera (Eggins et al., 2003; Hathorne et al., 2009). Like Mg/Ca, the values for Ba in crust carbonate are lower, which cannot be (solely) explained by migration to greater water depths during crust formation (Hathorne et al., 2009). This argues for an unknown additional imprint on Ba incorporation. On an intra-test scale, the distributions of Mg and Ba within the test wall of *Pulleniatina*



*obliquiloculata* have been shown to co-vary to some extent, with maximum concentrations
often, but not always, coinciding with the 'organic linings' (Kunioka et al., 2006). For some
other elements, including Mg and Sr, incorporation has been shown to be inter-dependent (e.g.
Mewes et al., 2015). This interdependency varies between pairs of elements and is explained
by a combination of simultaneous fractionation by the same process (e.g. Langer et al., 2016)
and by involvement of different processes during calcification (Nehrke et al., 2013). These
models and experimental results may imply that also the incorporation of Ba could be
influenced by these physiological processes and/ or the same fractionation process during
calcite precipitation (e.g. through lattice distortion; Mucci and Morse, 1983; Mewes et al.,

2015).

So far, Ba/Ca values have been reported for planktonic (Boyle, 1981; Lea and

Boyle, 1991; Lea and Spero, 1992; 1994; Hönisch et al., 2011; Marr et al., 2013; Hoffmann et
al., 2014) and low-Mg benthic species (Lea, 1995; Lea and Boyle, 1989; 1990; 1993; Reichart
et al., 2003). Although Mg/Ca is known to vary greatly between (benthic) foraminiferal species
(between ~1 and ~150 mmol/mol; Toyofuku et al., 2000; Bentov and Erez, 2006; Wit et al.,
2012) it remains to be investigated whether Ba/Ca also varies between benthic species with
different Mg/Ca. Ba/Ca in the planktonic species may be used to reconstruct (changes in) open
ocean alkalinity (Lea, 1995) and those published for benthics are more suitable to reconstruct
salinity in coastal and shelf seas (Weldeab et al., 2007; 2014; Bahr et al., 2013). The range in
Mg/Ca is known particularly for benthic foraminifera (e.g. Toyofuku et al., 2011; Sadekov et
al., 2014) and inter-species variability in Ba incorporation may therefore particularly hamper
application of (benthic) foraminiferal Ba/Ca. Here we present results from culture studies using
the larger benthic foraminifera, *Amphistegina lessonii* and *Heterostegina depressa*, two species
with different Mg/Ca (~50 mmol/mol; Segev and Erez, 2006 and ~120 mmol/mol; Dueñas-
Bohórquez et al., 2011, respectively). These results are compared to Ba/Ca in these species





from field samples. Together, calibration of Ba/Ca in these species against seawater Ba/Ca
allows evaluation and application of incorporated Ba across a wider range of foraminiferal taxa,
with contrasting element composition of their shell.

**2   Methods**
*2.1 Culture media*
To determine Ba/Ca partitioning, benthic foraminiferal culture experiments were set up with
five different seawater Ba/Ca ratios (54-92 µmol/mol). Media were prepared by increasing
$[Ba^{2+}]_{sw}$ while keeping the $[Ca^{2+}]_{sw}$ constant. The range of $[Ba^{2+}]$ used in these experiments
exceeds the range of concentrations found naturally and allows testing the applicability of
partition coefficients under conditions with artificially high seawater Ba/Ca. Seawater is only
slightly undersaturated with respect to barite ($BaSO_4$) and an increase in $[Ba^{2+}]$ in the sea water
will cause barite precipitation (Langer et al., 2009). To be able to increase $[Ba^{2+}]$ beyond its
natural range, artificial seawater was prepared with lower sulphate contents. All other salts were
added according to the recipe of Kester et al. (1967). As *Amphistegina lessonii* and
*Heterostegina depressa* do not grow well in 100% artificial seawater, the prepared media were
mixed with natural seawater in a ratio 9:1 (Mewes et al., 2014). To double check concentrations
and determine potential loss of elements due to precipitation, sorption and/or scavenging,
element concentrations of the culture media were determined by ICP-OES at the Alfred-
Wegener-Institute in Bremerhaven, except for Ba which was measured by ICP-MS at Utrecht
University (Table 1).
Culture media pH was adjusted to 8.0 by adding NaOH (1 M) to the prepared media. Before
the start of the experiments, dissolved inorganic carbon (DIC) and total alkalinity were
measured at the Alfred-Wegener-Institute. DIC was measured photometrically in triplicates
with a TRAACS CS800 QuAAtro autoanalyser with an average reproducibility of ± 10 µmol





L$^{-1}$. Alkalinity was calculated from linear Gran plots (Gran, 1952) after triplicate potentiometric
titration (Bradshaw et al., 1981) using a TitroLine aplpha plus auto sampler (Schott
Instruments). Parameters of the total carbonate system were calculated from temperature,
salinity, DIC and alkalinity using the program CO2SYS (Lewis and Wallace, 1998) adapted to
Excel by (Pierrot et al., 2006). The equilibrium constants K1 and K2 from Mehrbach et al.
(1973), as reformulated by Dickson and Millero (1987) were used (Table 1).

*2.2 Foraminiferal culturing*

Living specimens of *A. lessonii* and *H. depressa* were isolated from sediment collected at the
tropical aquarium of Burger's Zoo (Arnhem, The Netherlands) in August 2012 and transferred
to the Alfred-Wegener-Institute for the culture experiments. Healthy individuals of *A. lessonii*
showing pseudopodial activity, a dark brown cytoplasm and minimal signs of bleaching were
handpicked with a small brush under a Zeiss Stereo microscope and transferred to well plates.
Adult specimens of *H. depressa* were picked directly from the aquarium with soft tweezers.
After two weeks several individuals of both species underwent asexual reproduction. Individual
*H. depressa* parent cells produced sufficient numbers of juveniles to study separate clone
groups. Approximately 20 juveniles with two or three chambers from the same parent were
selected for every treatment and divided over two Petri dishes (diameter 55 mm). In total, two
clone groups were used in the experiments resulting in a total of at least 40 individuals per
treatment. Specimens of *A. lessonii* did not produce sufficient numbers of juveniles for analysis
of separate clone groups. Therefore, approximately 60 juveniles with two or three chambers
from different parents were selected per treatment and distributed evenly over three Petri dishes.
All experiments were carried out in an adjustable incubator (RUMED Rubarth Aparate GmbH)
at a constant temperature of 25 °C. As both species are symbiont-bearing, a 12:12 light:dark
cycle was applied with a constant photon flux density of approximately 250 μmol photons m$^-$





$^2s^{-1}$ during light hours. Pictures were taken weekly under a Zeiss Axiovert 200M inverted
microscope and maximal diameters of the shells were measured with the AxioVision software
to allow determining the chamber addition rates of the foraminifera in the experiments. The
experiments were terminated after six weeks.
All specimens were fed *Dunaliella salina* algae every three to four days. Although *A. lessonii*
hosts symbionts, this foraminiferal species does not exclusively rely on nutrients from their
symbionts, but also ingests algae (Lee, 2006). To avoid changes in the barium concentration of
the culture media, foraminifera were diluted as little as possible by the solution containing the
food for the foraminifera. For this purpose, foraminifera were fed 50 µl of a solution containing
algae that was centrifuged at 2000 rpm for 10 minutes. Algae concentrated at the bottom of the
tube were transferred to an empty tube with a pipette. To prevent changes in the culture media's
carbonate chemistry by algal photosynthesis the algae were killed by heating the concentrated
solution in an oven at 90 °C for 10 minutes. The cultures were transferred to new Petri dishes
every week to avoid excessive bacterial growth, potential build-up of waste products and
shortage of ions or nutrients. To prevent changes in salinity by evaporation media were
refreshed three days after the cultures were transferred to new dishes by pipetting approximately
5 ml of the old media out of the Petri dish and replacing it with the same volume of media from
the prepared batch.

*2.3 Sample preparation and analysis*

At the end of the culture experiment, specimens were cleaned by placing them in a 7% NaOCl
solution for approximately 30 minutes until completely bleached and organic material was
removed from the tests. This cleaning method is shown to have a similar impact on average
foraminiferal Ba/Ca values as cleaning with $H_2O_2$ (Pak et al., 2004). Specimens were then
rinsed three times for approximately 60 seconds in de-ionized water to remove the NaOCl and





any residual salts from the culture solutions. Cleaned foraminifera were put in an oven at 42 °C
until completely dry and mounted on sample holders using double sided adhesive tape.
Element composition of the calcite was determined using Laser Ablation-Inductively Coupled
Plasma-Mass Spectrometry (LA-ICP-MS) at Utrecht University (Reichart et al., 2003).
Monitored masses included $^{23}$Na, $^{24}$Mg, $^{26}$Mg, $^{27}$Al, $^{43}$Ca, $^{44}$Ca, $^{55}$Mn, $^{88}$Sr, $^{138}$Ba and $^{238}$U and
calibration was performed using a glass standard (NIST 610) that was ablated three times after
every 10-12 foraminiferal samples. Diameter of the ablation crater was set to 80 μm for all
specimens and pulse repetition rate was 6 Hz. The ablated calcite was measured and integrated
with respect to time. Energy density for the glass was higher than for the foraminifera (5 J/cm$^2$
and 1 J/cm$^2$, respectively). Although the resulting difference in ablation characteristics is not
likely to affect obtained foraminiferal element concentrations (Hathorne et al., 2008),
foraminiferal element concentrations were compared to those from an in-house made calcite
standard with known element concentrations and ablated at the same energy density as the
foraminifera (Dueñas-Bohórquez et al., 2009). Due to the lamellar nature of Rotallid
foraminifera, the final chamber was generally too thin for reliable determination of element/Ca
ratios. Therefore, the F-1 chamber of *A. lessonii* was ablated for every specimen. For *H.*
*depressa*, walls of the final two chambers were commonly too thin for reliable chemical results
and, therefore, the F-2 chamber was analysed. For each species, the final 6-7 chambers of ten
sufficiently large specimens were ablated to analyse intra-specimen variability in Ba/Ca and to
detect potential ontogenetic trends in Ba incorporation.

Elemental concentrations were calculated from the ablation profiles with the Glitter software,
using $^{43}$Ca as internal standard and values from Jochum et al. (2011) for concentrations of
elements in the NIST 610. This program integrates the ablation signal after subtracting the
background signal to calculate the elemental concentrations. To avoid contaminated intervals



of the ablation profile, sections with high $^{27}$Al and $^{55}$Mn counts were excluded from the analysis.
Ablation profiles with a duration shorter than 5 seconds were rejected as such short profiles are
unreliable due to poor counting statistics. Nine out of 188 ablation profiles were rejected for *A.*
*lessonii* and 7 out of 140 profiles from *H. depressa* were discarded, which is less than 5%.

*2.4 Field samples*

To compare the results from cultured specimens with Ba/Ca from specimens derived from
'naturel conditions', a number of living specimens of both *A. lessonii* and *H. depressa* were
isolated from the Zoo's stock and cleaned and prepared for LA-ICP-MS analyses as described
in 2.3. From both species, 7 specimens were ablated twice at the royal NIOZ using a
NWR193UC (New Wave Research) laser, containing an ArF Excimer laser (Existar) with deep
UV 193 nm wavelength and <4 ns pulse duration. Provided that the same reference material is
used, the use of multiple laser systems (see above) is shown not to bias obtained foraminiferal
element/Ca ratios (De Nooijer et al., 2014a). Laser ablation was performed with an energy
density of 1 J/cm$^2$ at a repetition rate of 6 Hz for calcite samples and an energy density of 5
J/cm$^2$ for the glass (NIST) standards. Helium was used as a carrier gas with a flow rate of 0.8
L/min for cell gas and 0.3 L/min for cup gas. From the laser chamber to the quadrupole ICP-
MS (iCAP-Q, Thermo Scientific), the He flow was mixed with ~0.4 L/min nebulizer Ar. Before
measuring the samples, the nebulizer gas, extraction lens, CCT focus lens and torch position
were automatically tuned for the highest sensitivity of $^{25}$Mg by laser ablating MACS-3. The
masses measured by the ICP-MS were $^{23}$Na, $^{24}$Mg, $^{25}$Mg, $^{27}$Al, $^{43}$Ca, $^{44}$Ca, $^{88}$Sr and $^{138}$Ba. JCp-
1, MACS-3 and an in-house (foraminiferal) calcite standard (NFHS) were used for quality
control and measured every 10 foraminiferal samples. Internal reproducibility of the analyses
was all better than 9%, based on the three different carbonate standards used. Intensity data
were integrated, background subtracted, standardized internally to $^{43}$Ca and calibrated against



the MACS-3 signal using a costume-built MATLAB routine within the program SILLS
(Guillong et al., 2008). Since ablation of the NIST SRM 610 and NIST SRM 612 could increase
the sodium background, they were only ablated and analyzed at the end of every sequence and
cones were cleaned before the next sequence. Accuracy of the analyses was better than 3%,
based on comparing the carbonate standards with internationally reported values (Okai et al.,
2002, Wilson et al., 2008). Signals were screened for surface contamination and parts of the
outside or inside of the shell with elevated Mg, Mn or Al values were eliminated from the area
selected for integration.
Seawater samples from the Zoo's aquarium were measured in duplicate using a sector field-
ICP-MS (Element2, Thermo Scientific). The ICP-MS was run in low resolution mode (24
cycles) for $^{138}Ba$ and in medium resolution (24 cycles) for $^{43}Ca$. Calibration was performed
through an external calibration series with increasing concentrations of Ba.

**3    Results**

*3.1 Test diameter increase*

Average shell diameters increased considerably during the experimental period (Figure 1).
Overall, increase in shell diameter did not significantly differ between treatments. Treatment C
(seawater Ba/Ca = 64 μmol/mol) for *A. lessonii*, however, shows somewhat reduced chamber
addition rates per incubated specimen. This may be the consequence of slightly higher mortality
under these conditions and a relatively high number of specimens that did not add any
chambers. Although not systematically investigated, 2 Petri dishes from this treatment
contained relatively many bleached (i.e. devoid of symbionts) specimens at the end of the 6-
week period.

*3.2 Barium incorporation*





Calcite Ba/Ca increases linearly with seawater Ba/Ca for both species (Figure 2; Table 2).
ANOVA performed on the individual data points combined with regression analyses reveals a
significant increase of $Ba/Ca_{cc}$ with $Ba/Ca_{sw}$ for both species (Table 3). Calculated regression
slopes result in a $D_{Ba}$ of 0.326 (±0.005) for *A. lessonii* and 0.777 (±0.007) for *H. depressa*
(Figure 3, solid lines). Regression lines are forced through zero as it seems reasonable to assume
that no Ba is incorporated into calcite when the Ba concentration in the seawater is zero.
Without this forcing, calculated partition coefficients would be 0.335 (±0.022) for *A. lessonii*
and 0.919 (±0.030) for *H. depressa*. The resulting partition coefficients (($Ba/Ca_{cc}$)/($Ba/Ca_{sw}$))
are constant and significantly different between the species (ANOVA) (~0.3 for *A. lessonii* and
~0.8 for *H. depressa*) over the range of seawater Ba/Ca studied here. The regression line for
$Ba/Ca_{cc}$ as a function of $Ba/Ca_{sw}$ for *A. lessonii* corresponds well with that reported for a number
of different low Mg species (Lea and Boyle, 1989).
The aquarium-derived specimens (field samples) had an diameter ranging from 550 to 1180 µm
(with an average of 975 µm) for *A. lessonii* and from 1380 to 2340 µm (average: 1936 µm) for
*H. depressa*. They had an average Ba/Ca of 15.4 (±2.3 SD) µmol/mol for *A. lessonii* and 35.7
(±14 SD) µmol/mol for *H. depressa*. In combination with the measured aquarium's seawater
Ba/Ca of 35.7 (±3.9 SD) µmol/mol, the partition coefficients for Ba vary between 0.43 and 1.0
for *A. lessonii* and *H. depressa*, respectively. Since the conditions in which the specimens from
the aquarium were grown, are relatively poorly constrained, they are not used for the regression
analysis, but are included (Figure 2) to compare with the cultured specimens.

*3.3 Intrachamber variability in Ba/Ca*
From both species, 10 specimens were used to quantify the relation between ontogeny (i.e. size-
dependent) and Ba incorporation into foraminiferal calcite. For this purpose, the final 6-7
chambers of these individuals were ablated (Figure 3). With the selected spot diameter (80 µm),



ablation of a small amount of material of adjacent chambers could not always be avoided. Some
chamber walls, particularly of the youngest (i.e. built latest) chambers, were too thin for reliable
measurements and were excluded from further analysis.
Since these specimens were cultured at different $Ba/Ca_{sw}$, the inter-chamber variability is
expressed as the difference of a single-chamber Ba/Ca and the individual's average Ba/Ca.
Positive single-chamber values indicate higher than average values, whereas negative values
indicate single-chamber Ba/Ca below that individual's average Ba/Ca (Figure 3).

In *H. depressa*, $Ba/Ca_{cc}$ increases significantly with subsequent chamber addition (Figure 3).
Regression analysis reveals an average decrease of 1.43 µmol/mol $Ba/Ca_{cc}$ per chamber (Table
4). $Ba/Ca_{cc}$ appears to decrease with chamber position in *A. lessonii,* although the ANOVA p-
value shows that this is statistically not significant. Still, removing one single outlier already
results in a p-value lower than 0.01. This implies that the current data set does not allow
rejecting the presence of a trend for *A. lessonii*.

*3.4 Relation between incorporation of barium and magnesium*
The species-specific single-chamber Mg/Ca and Ba/Ca combined for all treatments are
positively and significantly related (Figure 4). For *A. lessonii*, Mg/Ca = 3.1*Ba/Ca – 3.6 (t-
value = 12.2, p< 0.01 for the slope of the regression) and for *H. depressa*, Mg/Ca = 1.1*Ba/Ca
+ 92 (t-value = 14.8, p<0.01 for the slope). The slopes of these two regressions (3.1 and 1.1)
are significantly different: this is calculated by z = ($a_{Heterostegina}$ − $a_{Amphistegina}$)/ $\sqrt{(SE_{a,Heterostegina}^2 +}$
$SE_{a,Heterostegina}^2)$, where a is the value for the regression's slope and $SE_a$ is the slope's associated
standard error. For the slopes of the Mg/Ca-Ba/Ca regressions for *Amphistegina* and
*Heterostegina*, the resulting in a z-score is higher than >7, indicating that the two slopes are



significantly different. When combining the data from both species, the regression is
represented by: $Mg/Ca = 2.5*Ba/Ca +16$ (t-value = 31.4, p<0.01 for the slope).
When comparing the single-chamber $D_{Ba}$ with $D_{Mg}$, of all data combined, the partition
coefficient for Mg is over 30 times lower than that of for Ba (Figure 4). Over the range in
$Ba/Ca_{sw}$ studied here, the relation between $D_{Ba}$ and $D_{Mg}$ is linear within both species. For *A.*
*lessonii*, $D_{Mg} = 40*D_{Ba} - 2.0$ (t-value = 7.3, p< 0.01 for the slope of the regression) and for *H.*
*depressa*, $D_{Mg} = 29*D_{Ba} + 3.8$ (t-value = 6.5, p<0.01 for the slope). The slopes of these two
regressions (40 and 29) are not significantly different (z-score 1.6). When combining the data
from both species, the regression equals: $D_{Mg} = 34*D_{Ba} +0.073$ (t-value = 29.9, p<0.01 for the
slope.

**4   Discussion**
*4.1 Test diameter increase*
The range of Ba concentrations used in the experiments did not influence the increase in shell
diameter of either foraminiferal species (Figure 1). Compared to *H. depressa*, increases in shell
diameter (which is proportional to the chamber addition rate) for *A. lessonii* were slightly more
variable. To prevent barite precipitation it was necessary to reduce the sulphate concentration
below that typically measured in natural seawater. Sulphate concentrations between 0.1 and 1
mmol/L do not affect inorganic calcite growth (Reddy and Nancollas, 1976), but a decrease in
growth rates of approximately 30% was observed in coccolithophores growing in artificial
seawater with a sulphate concentration 10% that of natural seawater (Langer et al., 2009).
Although coccolithophores and foraminifera may respond differently to lowered sulphate
concentrations, this reduction could have hampered growth of the specimens in our culturing
experiment. Chamber addition rates of *A. lessonii* in a culture set-up with a sulphate
concentrations similar to that of natural seawater (Mewes et al., 2014) were approximately 20%





higher than chamber addition rates observed in our experiments. Since these experiments were
not performed simultaneously using specimens from the same batch, it is not straight forward
to compare absolute rates and therefore the 20% difference cannot unambiguously be attributed
to sulphate concentration (Hoppe et al. 2011). Unfortunately no data exist on the effect of
reduced sulphate concentrations on the uptake of trace elements in foraminiferal calcite.
However Langer et al. (2009) demonstrated that sulphate limitation had no discernible effect
on Ba incorporation in coccolithophore calcite.

*4.2 Barium incorporation*
The variability in Ba/Ca between individual ablation craters is considerable, but the average
foraminiferal Ba/Ca shows a consistent relation with seawater Ba/Ca. This implies that the
observed variability is a reflection of the inhomogeneous distribution in the test and hence
filtered out when averaging. This is similar to the behavior for Mg and Sr (Sadekov et al., 2008;
Wit et al., 2012; De Nooijer et al., 2014a) and underscores the power of single-chamber
analyses. If present, inhomogeneity in test wall Ba/Ca in combination with different cross
section sampled during the ablation potentially account for the observed variability. This would
imply that although large differences are observed within a test wall, the average still reliably
reflects sea water concentration (this paper) and for Mg, still reflects seawater temperature
(Hathorne et al., 2009). Comparing within-specimen and between-specimen variability, De
Nooijer et al. (2014a) showed that within specimen variability does not account for the complete
observed variability for Mg/Ca in *Ammonia tepida*. This seems to be similar for Ba/Ca
(compare Figure 4 in this paper with Figure 5 from De Nooijer et al., 2014a), which would
mean that on average 20 chambers need to be analyzed to reach a 5% relative precision (De
Nooijer et al., 2014a). This is not limited by the analytical precision, but rather due to inherent





biological inter-chamber and inter-specimen variability. To reduce ontogenetic variability, a
narrow size fraction should be analyzed.
Incorporation of Ba in *H. depressa* shows a partitioning which is about 2.5 times higher than in
*A. lessonii*. Such a large offset of $D_{Ba}$ between benthic species has not been observed before.
Lea and Boyle (1989) found $D_{Ba}$ = 0.37 ± 0.06 for *Cibicidoides wuellerstorfi*, *Cibicidoides*
*kullenbergi* and *Uvigerina* spp. for a series of core tops, comparable to the partition coefficient
reported here for *A. lessonii* (0.33 ± 0.022; Figure 2). In contrast, partition coefficients for Ba
in planktonic foraminifera are roughly only twice as low as these benthic foraminiferal
partitioning coefficients (0.14-0.19; Hönisch et al., 2011; Lea and Boyle, 1991; Lea and Spero,
1992). Although temperature, pH, salinity and pressure were initially proposed as potential
explanation for the offset between planktonic and benthic $D_{Ba}$ (Lea and Boyle, 1991; Lea and
Spero, 1992), studies by Lea and Spero (1994) and Hönisch et al. (2011) showed no significant
impact of temperature, pH and salinity on Ba incorporation into planktonic foraminiferal
calcite. This would leave hydrostatic pressure to explain the difference between benthic and
planktonic species. Our observations show, however, that the observed differences in $D_{Ba}$
between *H. depressa* and *A. lessonii* and also the offset with the planktonic species are inherent
to these species. There may be a small impact of environmental parameters, explaining the
slightly higher partition coefficients for Ba in the "field' specimens taken from the aquarium
compared to the cultured ones (Figure 2). The overall differences in partitioning seem to
coincide with different taxonomic groups, which may indicate that foraminifera may differ in
their controls on transporting ions from seawater to the site of calcification. For example, the
contribution of transmembrane transport versus that of seawater transport (i.e. leakage; Nehrke
et al., 2013 or vacuolization; Erez, 2003) may vary between species and thereby account for
differences in Mg/Ca, Ba/Ca, etc. (Nehrke et al., 2013).






### 4.3 Inter-chamber variability of Ba/Ca$_{cc}$

In both species cultured here, Ba/Ca$_{cc}$ decreases significantly from largest (i.e. built latest in
life) towards the smaller chambers (Figure 3). Observed trends were not significantly different
between *A. lessonii* and *H. depressa*, suggesting that Ba/Ca$_{cc}$ decreases at the same rate with
size, despite the overall difference in Ba/Ca$_{cc}$ (Figure 3). Since we always analyzed chambers
at the same position (F-1 for *A. lessonii* and F-2 for *H. depressa*) and since the final size of the
cultured specimens was similar between treatments (Figure 1), ontogenetic trends in Ba/Ca do
not influence the trends in Ba/Ca between treatments (Figure 2). Several other studies showed
that element/Ca ratios can vary with chamber position. Raitzsch et al. (2011), for example,
reported increasing B/Ca and decreasing Mg/Ca towards younger chambers in the benthic
*Planulina wuellerstorfi*. Such patterns maybe related to changes in the surface-to-volume ratio
or relative changes "vital effects" as foraminifera grow larger. For example, pH reduction in
the foraminiferal microenvironment is related to the specimen's size (Glas et al., 2012) and may
thereby affect the chemical speciation of minor and trace element, which in turn, may determine
their uptake rates. Hönisch et al. (2011), however, showed that seawater pH has no noticeable
effect on Ba incorporation in planktonic foraminiferal calcite, rendering changes in the pH of
the foraminiferal microenvironment an unlikely explanation to account for the observed
chamber-to-chamber variability in Ba/Ca. Alternatively, changes in the metabolic rate, the
instantaneous calcification rate, or a different partitioning between the impacts of the life
processes may lead to the observed ontogenetic trend.
Bentov and Erez (2006) argued that decreasing Mg/Ca with foraminifera test size could be
explained by relatively high Mg-concentrations at or near the primary organic sheet (POS),
which is the organic matrix on which the first layer of calcite precipitates during the formation
of a new chamber. With the formation of a new chamber, a low-Mg calcite layer is deposited





over all existing chambers, so that the high-Mg phase is being 'diluted' as more layers are
deposited (Bentov and Erez, 2006). Future studies may indicate whether Ba/Ca is also
heterogeneously distributed within chamber walls, by for example, being enriched close to the
POS (Kunioka et al., 2006). If this is the case, lamellar calcification mode may also result in
changing Ba/Ca with chamber position.

*4.4 Coupled incorporation of barium and magnesium*
If incorporation of Ba and Mg (and Na, Sr and B) are physically linked during
biomineralization, inter-species differences in composition may likely be correlated across the
various elements. The correlation between Mg/Ca and Ba/Ca within and between species
(Figure 4) suggests that these two elements are simultaneously affected during their
incorporation. The relationship between Mg/Ca and Ba/Ca is different between the two species,
which may be (partly) caused by the variability in seawater chemistry between treatments (i.e.
seawater Ba/Ca and Mg/Ca; Table 1). Alternatively, incorporation of Mg in *H. depressa* may
be close to the maximum concentration of Mg that can be incorporated into a calcite crystal
lattice at ambient conditions (Morse et al., 2007). This may result in an overall asymptotic
relationship between Mg/Ca and Ba/Ca as Mg/Ca approaches ~200 mmol/mol (Figure 4).
When correcting for the different seawater Ba/Ca and Mg/Ca between treatments, incorporated
Ba and Mg correlate similarly within, as well as, between the two species studied here (Figure
4). This suggests that these elements are coupled during biomineralization and that the ratio of
Ba and Mg in seawater is preserved during calcification by these species of foraminifera. When
comparing the relation between Ba/Ca and Mg/Ca from other benthic species (e.g. Lea and
Boyle, 1989; figure 2; more refs), the coupling between Ba- and Mg-incorporation is likely
similar across a wide range of benthic foraminiferal species.



*4.5 Biomineralization and element incorporation*

Foraminiferal biomineralization determines incorporation of many elements and fractionation
of many isotopes during the production of new chamber as indicated by overall large
compositional differences between inorganically precipitated and foraminiferal calcite (Erez,
2003; Bentov and Erez, 2006; Nehrke et al., 2013; De Nooijer et al., 2014b). For example,
Mg/Ca ratios in many species are orders of magnitude lower than what is expected from
inorganic precipitation experiments. Additionally, Mg/Ca varies considerably between
foraminiferal species and especially between species known to have different calcification
strategies (Bentov and Erez, 2006; Toyofuku et al., 2011; Wit et al., 2012; De Nooijer et al.,
2009; 2014b). Other elements such as Sr (e.g. Elderfield et al., 2000) and B/Ca (e.g. Allen et
al., 2012) also vary significantly between species. Generally, concentrations for these elements
correlate within taxa and hence species incorporating relatively much Mg, also have high (for
example) Sr/Ca, B/Ca and Na/Ca. Miliolids and many 'Large Benthic Foraminifera' (LBF)
produce calcite with Mg/Ca up to 100-150 mmol/mol (Toyofuku et al., 2000; Dueñas-
Bohórquez et al., 2011; Sadekov et al., 2014; Evans et al., 2015), while most planktonic and
symbiont-barren benthic foraminifera produce test calcite with Mg/Ca values ranging from 1-
10 mmol/mol (e.g. Nürnberg et al., 1996; Elderfield et al., 2002; Lear et al., 2010; Wit et al.,
2012; De Nooijer et al., 2014b). The same distinction is observed for B/Ca (compare e.g. Allen
et al., 2012 and Kazcmarek et al., 2015), Li/Ca (Lear et al., 2010 versus Evans et al., 2015),
Na/Ca (Wit et al., 2013 versus Evans et al., 2015) and Sr/Ca (e.g. Dueñas-Bohórquez et al.,
2011). The correlation between relatively high (for example) Mg/Ca, Sr/Ca and B/Ca
corresponds to the observed trends in the data presented here for Ba/Ca and Mg/Ca in *H.*
*depressa* and *A. lessonii* (Figure 4). The Mg/Ca in the former species is approximately 2.5 times
that of the latter, which is similar to the difference in Ba/Ca ratios between these species and
implies that Ba changes in concert with Mg, which is consistent with the single-chamber



correlation between Mg/Ca and Ba/Ca (Figure 4). Although such a change could potentially be
caused inorganically by differences in Mg opening up the crystal lattice in such a way that it
can accommodate more or less Ba. Such a mechanism is described for Mg and Sr (e.g. Morse
and Bender, 1990; Mucci and Morse, 1983; Mewes et al., 2015; Langer et al., 2016) and may
also apply to Ba incorporation and the influence of Mg ions that increase stress in the calcite
crystal lattice. However, such a mechanism is unlikely a sufficient explanation, since it is
observed that Ba incorporation does not change in the planktonic *Orbulina universa* cultured at
temperatures from 18 and 26 $^{0}$C (Hönisch et al., 2011), even though this temperature range
roughly doubles the shell's Mg/Ca (Lea et al., 1999). Unless the strain of incorporated Mg ions
does not increase linearly with its concentration, the covariance between Mg and in this case
Ba may well be interrelated during an earlier stage of the biomineralization process, e.g. during
their transport from the surrounding seawater into the site of calcification (Erez, 2003; De
Nooijer et al., 2014b).
Interestingly, the partitioning of different elements is not the same between taxa. For example,
Sr/Ca in LBFs is approximately twice as high (Dueñas-Bohorquez et al., 2011; Evans et al.,
2015) as in planktonic species (Elderfield et al., 2002; Dueñas-Bohórquez et al., 2009; Hendry
et al., 2009), whereas the ratio between the $D_{Mg}$ of these groups is between 10 and 100 (see
above). Comparing the offset of D between groups as a function of D itself shows an
approximate logarithmic correlation (Figure 5). The distinction between the two groups on basis
of their element signature coincides with known differences in biomineralization controls.
Element controls in low-Mg species are thought to be determined by (highly) selective trans-
membrane ion transporters, (limited) leakage of seawater into the site of calcification and/or
selective $Mg^{2+}$-removal (Nehrek et al., 2013; De Nooijer et al., 2014b). Miliolid foraminifera
belong to the high-Mg foraminiferal group and are known to secrete their calcite intracellularly
(Hemleben et al., 1986; Ter Kuile and Erez, 1991; De Nooijer et al., 2009). These intracellular





vesicles may be derived from endocytosed seawater and therefore contain relatively high
concentrations of $Mg^{2+}$, $Ba^{2+}$ and other ions present in seawater, although so far only Sr/Ca and
Mg/Ca of Miliolid foraminifera have been published (supplementary information). The
biomineralization of non-Miliolid, intermediate- and high-Mg benthic foraminifera may
employ characteristics of both these types of calcification and therefore incorporate moderately
to high concentrations of elements (cf Segev and Erez, 2006).

*5*    **Conclusions**
Results from this study indicate that differences in $D_{Ba}$ between species of foraminifera are
larger than previously thought. This implies that species-specific Ba partition coefficients need
to be applied to reconstruct past $Ba/Ca_{sw}$ and/or salinity (Lea and Boyle, 1989; Weldeab et al.,
2007). Moreover, our results underscore the necessity to account for size-related effects on
$Ba/Ca_{cc}$. This effect may bias obtained $Ba/Ca_{cc}$ particularly when using single chamber
measurements. When determining $Ba/Ca_{cc}$ by dissolution of whole shells, the contribution of
smaller chambers (with lower $Ba/Ca_{cc}$) is relatively small compared to a specimen's overall
Ba/Ca and thus does not affect average values. Our results also show that within species as well
as between species, single-chambered Mg/Ca and Ba/Ca are linearly correlated. The difference
in Ba/Ca between the two species studied here fits with previously observed variability in
element/Ca ratios between foraminifera taxa and may correspond to differences in their
biomineralization mechanisms.

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





**Tables**

*Table 1: measured concentrations of major and minor ions, temperature, salinity and carbonate chemistry in the five culture media (A-E).*

| Treatment | A | B | C | D | E |
|---|---|---|---|---|---|
| Ba (nmol/kg) | 488.5 | 535.5 | 611.0 | 608.4 | 854.6 |
| Ca (mmol/kg) | 9.1 | 9.5 | 9.6 | 9.2 | 9.3 |
| Ba/Ca$_{sw}$ (mmol/mol) | 53.68 | 56.36 | 63.64 | 66.14 | 91.89 |
| Na (mmol/kg) | 402 | 416 | 389 | 383 | 384 |
| B (mmol/kg) | 11 | 11 | 12 | 11 | 11 |
| K(mmol/kg) | 0.40 | 0.46 | 0.43 | 0.43 | 0.42 |
| Mg (mmol/kg) | 55 | 58 | 59 | 53 | 53 |
| Sr (mmol/kg) | 0.11 | 0.11 | 0.12 | 0.11 | 0.11 |
| Mg/Ca$_{sw}$ (mol/mol) | 6.04 | 6.11 | 6.15 | 5.76 | 5.70 |
| T (°C) | 25 | 25 | 25 | 25 | 25 |
| Salinity | 32.4 | 32.4 | 32.4 | 32.4 | 32.4 |
| TA (µmol/kg) | 2445 | 2450 | 2662 | 2437 | 2429 |
| DIC (µmol/kg) | 2244 ± 3 | 2246 ± 6 | 2464 ± 7 | 2236 ± 7 | 2228 ± 9 |
| $\Omega_{calcite}$ | 3.9 | 3.9 | 4.0 | 3.9 | 3.9 |

*Table 2. Measured Ba/Ca and Mg/Ca for* A. lessonii *and* H. depressa *for each treatment.*

| Treatment | A | B | C | D | E |
|---|---|---|---|---|---|
| *A. lessonii* | | | | | |
| n | 40 | 43 | 17 | 36 | 43 |
| Ba/Ca (µmol/mol) | 15.8 | 19.6 | 18.8 | 22.9 | 29.9 |
| SD | 3.3 | 3.6 | 3.0 | 4.5 | 5.5 |



| Mg/Ca (mmol/mol) | 37.9 | 49.2 | 70.1 | 89.6 | 80.4 |
|---|---|---|---|---|---|
| SD | 10 | 13 | 19 | 33 | 29 |
| *H. depressa* | | | | | |
| n | 26 | 27 | 23 | 25 | 32 |
| Ba/Ca (µmol/mol) | 41.1 | 41.5 | 46.0 | 50.8 | 74.9 |
| SD | 6.2 | 4.3 | 3.9 | 5.7 | 3.9 |
| Mg/Ca (mmol/mol) | 150 | 135 | 123 | 168 | 177 |
| SD | 12 | 11 | 6 | 29 | 7 |


*Table 3. Parameters of the regression analysis and ANOVA tests for significance of the*
*regression. Both average Ba/Ca$_{cc}$ of each experimental condition (n=5) and all chamber-*
*specific Ba/Ca$_{cc}$ (n=133/ 179) were tested versus the Ba/Ca of the 5 treatments.*

| | | | Regression analysis | ANOVA | |
|---|---|---|---|---|---|
| Parameter | Species | n | $R^2$ | F-value | p-value |
| Ba/Ca$_{sw}$ vs Ba/Ca$_{cc}$ | *H. depressa* | 133 | 0.88 | 940 | <0.01 |
| | *A. lessonii* | 179 | 0.56 | 227 | <0.01 |
| Ba/Ca$_{sw}$ vs average Ba/Ca$_{cc}$ | *H. depressa* | 5 | 0.99 | 247 | <0.01 |
| | *A. lessonii* | 5 | 0.91 | 32 | 0.011 |


*Table 4. ANOVA parameters of single-chamber measurements*

| ANOVA | Species | F | p |
|---|---|---|---|
| | *A. lessonii* | 2.47 | 0.06 |
| | *A. lessonii (f-1 and f-2)* | 0.11 | 0.744 |
| | *H. depressa* | 6.09 | < 0.01 |





**Figures**

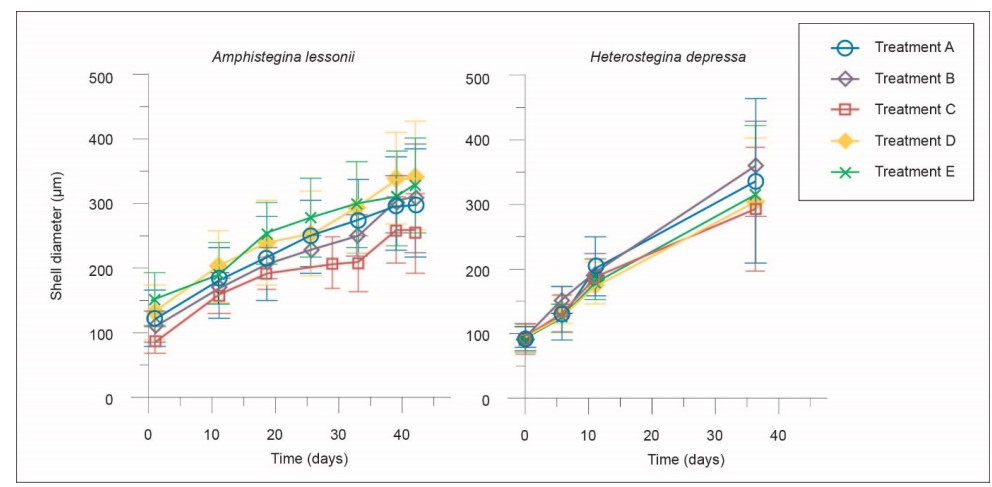


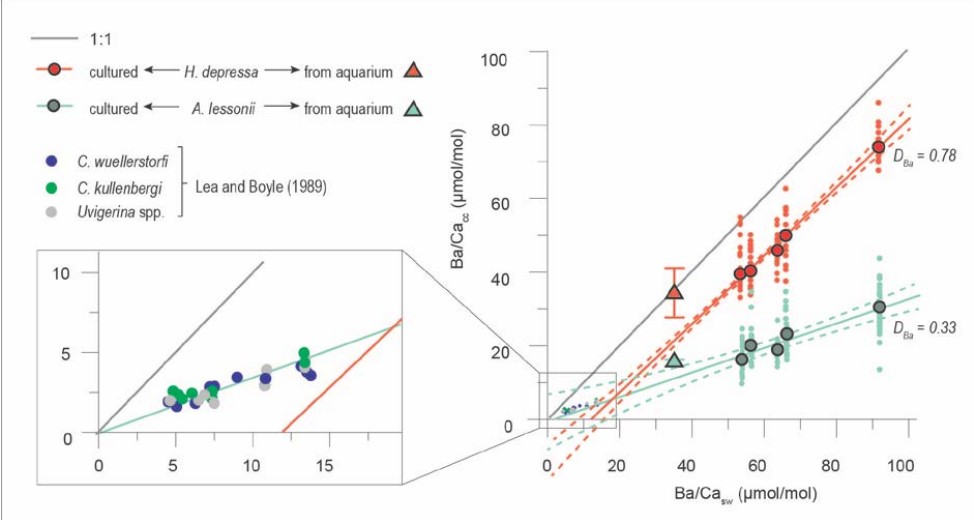









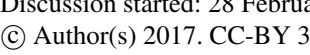



**Figure Captions**

*Figure 1. Average increase in shell diameter for* A. lessonii *(left panel) and* H. depressa *(right panel). Dots represent the average of all analysed individuals from one treatment. Error bars represent the standard deviation of the mean.*

*Figure 2. Foraminiferal Ba/Ca as a function of seawater Ba/Ca. Light circles indicate individual laser ablation measurements, larger, darker shaded circles represent the average Ba/Ca$_{cc}$ for one treatment. Relative standard deviation varies between 16 and 20% for Ba/Ca$_{cc}$ in* A. lessonii *and between 5 and 15% for* H. depressa. *Average Ba/Ca for the two species collected from the aquarium are indicated by triangles (+/- 1 SD) and were not taken into account when calculating the regression. Calculated regressions are accompanied by their 95% confidence intervals (dashed lines) over the Ba/Ca$_{sw}$ range from 50 to 90 µmol/mol. Data from Lea and Boyle (1989) is plotted additionally for comparison.*

*Figure 3. Average (large, darker shaded circles) and single chamber measurements (lighter circles) Ba/Ca$_{cc}$, expressed as their deviation from the mean shell Ba/Ca$_{cc}$ for* A. lessonii *(left) and* H. depressa. *Error bars represent the standard deviation of the mean, the dashed lines in the right panel indicate the 95% confidence intervals for the linear regression.*

*Figure 4. Relation between the Ba/Ca and Mg/Ca (left panel) and the partition coefficients for Ba and Mg (right panel). Every dot represents one single-chamber measurement. The data for* A. lessonii *are indicated by circles, those for* H. depressa *are represented by diamonds. Every treatment (A-E; Table 1) is indicated by a separate color.*



*Figure 5: Partition coefficients for Li, B, Na, Mg, Sr and Ba for two groups of foraminifera*
*(Large Benthic Foraminifera+Miliolids and the low-Mg species). Data on which the average*
*partition coefficients are based, are listed in the online supplement, the ranges indicate the*
*maximum range in published partition coefficients. The linear regression between the partition*
*coefficients for these two groups is described by: $D_{plankton/low\ Mg-benthic}=0.3992*D_{miliolid/LBF} +$*
*0.0081. Elemental results for Milliolid species are confined to Mg/Ca and Sr/Ca. Li/Ca ratios*
*were taken from Delaney et al. (1985), Hall and Chan (2004a), Marriott et al. (2004), Yu et al.*
*(2005), Ni et al. (2007), Bryan and Marchitto (2008), Hathorne et al. (2009), Dawber and*
*Tripati (2012) and Evans et al. (2015); B/Ca ratios are from Yu et al. (2005), Yu and Elderfield*
*(2007), Foster (2008), Hendry et al. (2009), Allen et al. (2011; 2012), Dawber and Tripati*
*(2012), Babila et al. (2014) and Kaczmarek et al. (2015); Na/Ca are from Delaney et al. (1985),*
*Ni et al. (2007), Bian et al. (2009), Wit et al. (2013) and Evans et al. (2015); Mg/Ca are from*
*Toyofuku et al. (2000), Raja et al. (2005), Yu et al. (2005), Elderfield et al. (2006), Segev and*
*Erez (2006), Hendry et al. (2009), Dueñas-Bohórquez et al. (2009; 2011), Dawber and Tripati*
*(2012), Wit et al. (2012; 2013), Babila et al. (2014), De Nooijer et al. (2014a), Sadekov et al.*
*(2014) and Evans et al. (2015). Foraminiferal Sr/Ca are taken from Raja et al. (2005), Yu et*
*al. (2005), Hendry et al. (2009), Dueñas-Bohórquez et al. (2009; 2011), Dawber and Tripati*
*(2012), Wit et al. (2013), De Nooijer et al. (2014a) and Evans et al. (2015). Ba/Ca are from*
*this study, Lea and Boyle (1989), Lea and Boyle (1991), Lea and Spero (1994), Hall and Chan*
*(2004b), Ni et al. (2007), Hönisch et al. (2011) and Evans et al. (2015).*