# Peer review of "Biogeosciences Discuss., doi:10.5194/bg-2017-45, 2017 Manuscript under review for journal Biogeosciences Discussion started: 28 February 2017"

_Biogeosciences, 2017_

## Referee Comment (RC1) · Anonymous Referee #1 · 26 Mar 2017

Authors report new and important results of Ba/Ca and Mg/Ca measurements using LA-ICPMS of two species of benthic foraminifera. The study is technically sound and show sufficient number of important findings. I highly recommend that this manuscript should be published. The followings are my minor comments that would be better to be considered before acceptance of this manuscript.

Line 40: the mechanism why Ba can be used as a proxy of total alkalinity and salinity should be explained more (Ba shows nutrient-type distribution in the ocean; Ba is rich in terrestrial water, etc.).

Lines 133-134: I could not understand this line "foraminifera were diluted as little as possible by the solution containing the food for the foraminifera." It is seawater that can be diluted, right?

[Figure]

Line 143: It would be nice if an explanation regarding "the batch seawater" appears in Sec. 2.1. Also, how much seawater was prepared in each treatment?

Line 154: It would be better to describe LA-ICPMS setups here (carrier gas, flow rate, ICPMS, laser type, etc.)

Lines 168-169: I wonder ten large specimens were randomly sampled among all the treatments?

Line 176: Weren't high Mg counts used for the data screening?

Line 184: What the Zoo's stock mean? Sediment?

Line 185: There is no explanation what "the royal NIOS" stands for.

Line 191: Please add "grass standards NIST 610 and 612" here.

Lines 230-231: The linear regression line of the data of H. depressa in Fig. 2 is not forced through zero. Is this intentionally or mistakenly?

Lines 243-245: I think these sentences need to be revised. I don't think the "field" data should be removed from the discussion, but there must be better explanation, like "the aquarium derived data is consistent with the culturing derived data, but it was not used in the regression analysis, since the conditions in which (...)".

Lines 262-263: I could not understand this line. Which data, do you mean, is the outlier?

Line 319: What the word "complete" mean?

Line 322 What is the significance or importance of "5% precision"?

Lines 324-325: Is this sentence implication for whole shell analysis in paleoceanography?

Lines 340-342: I could not read through this line. Do you mean "a small difference of environmental parameter may partly explain a slight difference in D_Ba between

aquarium and cultured samples"?

Fig. 4: A difference between symbols representing two species should be more distinct: For example, open and closed symbols. There seems no difference in a blurry and small figure.

―――――――――――――――――――

---

## Referee Comment (RC2) · Anonymous Referee #2 · 14 Apr 2017

The manuscript 'Ba incorporation in benthic foraminifera' presents laboratory culture barium data for two species, *Heterostegina depressa* and *Amphistegina lessonii*. Ba/Ca ratios in foraminifera may be used to trace past changes in seawater [Ba], which in turn may be related to (e.g.) salinity or alkalinity, and the proxy is therefore of broader community interest. Whilst Ba/Ca has been successfully applied as a proxy using low-Mg foraminifera for some time, this study provides seawater-shell Ba/Ca calibrations for two high-Mg species. I was missing an explanation in the text of why the authors chose to use a range of seawater Ba/Ca ratios that are much higher than natural seawater. Nonetheless, the data are of good quality, and are suitable for publication in *Biogeosciences*.

**1 Comments**

1. In the abstract and introduction there is no mention of upwelling, which may complicate the use of Ba/Ca as a salinity proxy, especially in benthic organisms.

2. Lines 64-67, line 327, and lines 455-456. This is not the first time that Ba/Ca has been investigated in a high-Mg benthic species and therefore these sentences should be rephrased. Evans *et al.* [2015] *GCA* report Ba/Ca data for the high-Mg species *Operculina ammonoides* under variable seawater [Ba], and found a barium distribution coefficient (0.66) similar to that reported here for *H. depressa*. van Dijk *et al.* [2017] also report Ba/Ca data for *H. antillarum* which has a distribution coefficient of 1.2-2.2 according to that study.

3. Section 2.2. Please state the approximate volume of seawater in which the cultures took place. As these cultures were performed in petri dishes, presumably the volume was relatively small? If so: (1) How was evaporation monitored and avoided? (2) It is likely that the foraminifera modified the carbonate chemistry of the seawater in between water exchanges (once per week). Was this monitored?

4. The phrasing of lines 148-149 implies that the cleaning procedure has an impact on measured Ba/Ca. Either rephrase or state what this impact is.

5. There is far more detail given for the laser-ablation performed at the royal NIOZ. Whilst a reference is provided for the system at Utrecht University, it may be useful to state which LA and ICPMS systems were used and the wavelength of the laser for easy comparison. What is the accuracy and precision of the system used at Utrecht University?

6. Lines 164-169. Here the authors state that the final chamber of *A. lessonii* and the final two chambers of *H. depressa* did not yield reliable data because the

walls are thin. However, Fig. 3 shows data for F and F-1 for both species. Is this a mistake? If not, these data should not be shown if they are not reliable.

7. Lines 175-176. What is the possible source of Al and Mn contamination in cultured foraminifera?

8. Section 2.4. Consider changing the phrase 'field specimens'.

9. Lines 230-231. I agree that it is reasonable to assume no barium incorporation when there is no barium in seawater. However, forcing a linear regression through the origin also assumes that seawater and shell Ba/Ca must be linearly related across the full range of seawater Ba/Ca ratios, which may not be the case. Consider that the *H. depressa* zoo aquarium sample may be in agreement with the cultures if the regression is not forced through the origin.

10. Lines 232-233. Technically, if the regressions are not forced through the origin then there is not a single partition coefficient value, I suggest this is rephrased in terms of the seawater-shell Ba/Ca slopes.

11. Lines 241-242. Why is the aquarium seawater Ba/Ca higher than most natural seawater? If this is known it would be useful to state the reason. What is the meaning of the sentence starting on line 243? Is it an analytical problem or is there reason to suspect the aquarium seawater Ba/Ca ratio was not constant?

12. Section 3.3. It would aid the interpretation of these interesting data if the reader had an idea of how much of the foraminifera the final five chambers represent. Approximately how many chambers were precipitated in culture? Consider adding a representative image.

13. Lines 259-260. The phrasing is confusing here. Ba/Ca increases in the first sentence but decreases in the second sentence. Rephrase for consistency.

14. Lines 273-276. If the two slopes are significantly different, why combine the data from both species?

15. Section 4.2 and Figure 4. I am surprised that the range in Mg/Ca is so large, both within and between experiments, and this requires further explanation. For example, compare these *H. depressa* data to those reported in Raitzsch *et al.* [2010]. In that paper the Mg/Ca 2SD was 17 to 24 mmol mol$^{-1}$ ($\sim$10-20%), which is comparable to other studies reporting laser-ablation data. Here, some experiments are in line with this while others have a far larger range, for example *H. depressa* treatment D ($\sim$110-190 mmol mol$^{-1}$). The *A. lessonii* data are even more surprising, treatment D has a range from 30-140 mmol mol$^{-1}$, and treatment C has a range of 30-120 mmol mol$^{-1}$. Why is there so much variation compared to other studies? Could something in the experimental design have resulted in this? Is it a result of using juvenile foraminifera?

16. Line 363-366. Alternatively, van Dijk *et al.* [2017] showed that $p$CO$_2$ does impact Ba incorporation in *Amphistegina*, so perhaps the microenvironment carbonate chemistry can help to explain these data.

17. Line 428-431. This argument is not valid, we would not expect a doubling in *O. universa* Mg/Ca to exert a resolvable impact on Ba or Sr incorporation. For example, the $D_{Sr}$-Mg/Ca slope for inorganic calcite is $9.1 \times 10^{-4}$ [Mucci & Morse, 1983], so that a change in shell Mg/Ca of 10 mmol mol$^{-1}$ would result in a Sr/Ca increase of just $\sim$0.1 mmol mol$^{-1}$. If the relationship between $D_{Ba}$ and Mg/Ca is similar, we would not observe this effect in *Orbulina*. It is visible in high-Mg species only because the shell Mg/Ca ratios are 1-2 order of magnitude higher than low-Mg foraminifera.

18. Line 446. I think repeating the assertion that miliolids calcify intracellularly should be avoided. As stated a few lines later, it is intracellular only in the sense that

calcification takes place from endocytosed seawater, which may well be the case for rotaliid foraminifera as well.

19. Line 449-450. van Dijk *et al.* [2017] report Na, Zn and Ba data for miliolid foraminifera.

20. Line 458. You could also reference Hoffmann *et al.* [2014] *Geology* **42**:579 and Evans *et al.* [2015] *G-cubed* **16**:2598.

21. Figure 3. It would be more intuitive to plot the final chamber on the right hand side of the graphs, so that time goes forward from left to right.

22. Figure 4. Please use symbols for the two species that are easier to distinguish. Consider plotting the slopes discussed in the text.

**2 Typos**

1. Line 183. 'naturel'.

2. Line 201. 'costume-built'.

3. Line 221. Write out '2'.

4. Line 274. Delete 'in a'.

5. Line 359. 'maybe'.

---

## Author Comment (AC1) · 9 Jun 2017

Dear Editor,

We are glad to have received two constructive sets of comments. We have addressed all of them to improve our manuscript and we would like you to consider the revised manuscript for publication in Biogeosciences. Below, we have repeated the reviews and added point-by-point our reply.

Sincerely,

Lennart de Nooijer, also on behalf of the other authors,

Reviewer #1

[Figure]

Authors report new and important results of Ba/Ca and Mg/Ca measurements using LA-ICPMS of two species of benthic foraminifera. The study is technically sound and show sufficient number of important findings. I highly recommend that this manuscript should be published. The followings are my minor comments that would be better to be considered before acceptance of this manuscript.

Line 40: the mechanism why Ba can be used as a proxy of total alkalinity and salinity should be explained more (Ba shows nutrient-type distribution in the ocean; Ba is rich in terrestrial water, etc.).

The first sentences of the abstract have been re-written and now explain why foraminiferal Ba/Ca can be used to reconstruct alkalinity and salinity (lines 39-47).

Lines 133-134: I could not understand this line "foraminifera were diluted as little as possible by the solution containing the food for the foraminifera." It is seawater that can be diluted, right?

This is now changed into: "...the water in the dishes containing the foraminifera was diluted as little as possible...".

Line 143: It would be nice if an explanation regarding "the batch seawater" appears in Sec. 2.1. Also, how much seawater was prepared in each treatment?

The concentrations of elements, temperature, salinity and carbonate chemistry from the 'batch seawater' are listed in Table 1. The recipe followed to make this stock solution is described in lines 101-105 and we included the total volume of seawater prepared (5 liters for each treatment; line 102).

Line 154: It would be better to describe LA-ICPMS setups here (carrier gas, flow rate, ICPMS, laser type, etc.)

We have added some more information about the employed setup and settings used in this section (lines 168-171).

Lines 168-169: I wonder ten large specimens were randomly sampled among all the treatments?

This is true indeed: for every treatment, 2 specimens (i.e. 10 in total) were randomly selected. This is now added to line 187.

Line 176: Weren't high Mg counts used for the data screening?

No, but since surface contamination is most clearly visible in elements like Mn and Al, and the elevated peaks in Mg usually coincide with peaks in these two elements, many of the excluded parts of the ablation profiles also contain relatively much Mg. This is exactly why these parts are excluded. This is now clarified in line 194-197.

Line 184: What the Zoo's stock mean? Sediment?

This means the sediment collected originally at Burgers' Zoo and now kept in our laboratory to isolate the specimens that were cultured at each of the five treatments (beginning of section 2.2). This is now indicated at lines 205-206.

Line 185: There is no explanation what "the royal NIOS" stands for.

Royal NIOZ is the affiliation of some of the authors and the laboratory at which these analyses were performed. This is mentioned here to underline the difference of the system employed for these measurements compared to that used for the cultured specimens (which were measured at Utrecht University).

Line 191: Please add "grass standards NIST 610 and 612" here.

We have added "610" to this sentence (line 213). NIST 612 was not used, by the way.

Lines 230-231: The linear regression line of the data of H. depressa in Fig. 2 is not forced through zero. Is this intentionally or mistakenly?

This is now corrected: see adjusted figure 2.

Lines 243-245: I think these sentences need to be revised. I don't think the "field" data

should be removed from the discussion, but there must be better explanation, like "the aquarium derived data is consistent with the culturing derived data, but it was not used in the regression analysis, since the conditions in which (...)".

We agree and have changed the text accordingly.

Lines 262-263: I could not understand this line. Which data, do you mean, is the outlier?

We have tried to clarify this sentence by rephrasing it so that it now reads "...lower than 0.01, indicating that the current data set does not allow rejecting the presence of a trend for A. lessonii." (lines 288-289).

Line 319: What the word "complete" mean?

We have changed this to "...does not account for all of the observed variability in Mg/Ca..." (lines 347-348 in the revised version of our manuscript).

Line 322 What is the significance or importance of "5% precision"?

This indicates that measuring 20 (or more) individual chambers will result in an estimate within +/- 5% of the true mean. "on average" is replaced by "at least" in this sentence.

Lines 324-325: Is this sentence implication for whole shell analysis in paleoceanography?

For example. This suggestion has been added to this sentence.

Lines 340-342: I could not read through this line. Do you mean "a small difference of environmental parameter may partly explain a slight difference in D_Ba between aquarium and cultured samples"?

This is indeed what we meant. This sentence is changed into: "A small impact of environmental parameters other than seawater Ba/Ca may account for the slightly higher

DBa in the "field' specimens taken from the aquarium compared to the cultured ones (Figure 2)."

Fig. 4: A difference between symbols representing two species should be more distinct: For example, open and closed symbols. There seems no difference in a blurry and small figure.

We have revised this figure: in addition to using different symbols (circle versus diamond), the diamonds are now open, whereas the circles have remained filled.

Reviewer #2 The manuscript 'Ba incorporation in benthic foraminifera' presents laboratory culture barium data for two species, Heterostegina depressa and Amphistegina lessonii. Ba/Ca ratios in foraminifera may be used to trace past changes in seawater [Ba], which in turn may be related to (e.g.) salinity or alkalinity, and the proxy is therefore of broader community interest. Whilst Ba/Ca has been successfully applied as a proxy using lowMg foraminifera for some time, this study provides seawater-shell Ba/Ca calibrations for two high-Mg species. I was missing an explanation in the text of why the authors chose to use a range of seawater Ba/Ca ratios that are much higher than natural seawater. Nonetheless, the data are of good quality, and are suitable for publication in Biogeosciences.

We thank the reviewer for his/her constructive comments. At the end of the introduction (lines 84-88) we have now added the rationale for using Ba/Ca exceeding the natural range. This extended range facilitates testing the mechanisms underlying Ba-incorporation in foraminiferal calcite (see section 4.5) and at the same time (given that the $Ba/Ca_{acc}$ responds linearly to increasing $Ba/Ca_{sw}$), decreases the uncertainty in the calibration over the paleoceanographically relevant range in seawater Ba/Ca.

Comments 1. In the abstract and introduction there is no mention of upwelling, which may complicate the use of Ba/Ca as a salinity proxy, especially in benthic organisms.

We have now added a sentence to the beginning of the introduction about the influence of upwelling to surface water Ba/Ca (lines 47-48). "These reconstructions can be complicated by upwelling affecting surface Ba/Ca (Lea et al., 1989; Hatch et al., 2013)."

2. Lines 64-67, line 327, and lines 455-456. This is not the first time that Ba/Ca has been investigated in a high-Mg benthic species and therefore these sentences should be rephrased. Evans et al. [2015] GCA report Ba/Ca data for the highMg species Operculina ammonoides under variable seawater [Ba], and found a barium distribution coefficient (0.66) similar to that reported here for H. depressa. van Dijk et al. [2017] also report Ba/Ca data for H. antillarum which has a distribution coefficient of 1.2-2.2 according to that study.

In the introduction (now lines 73-74), we added "Ba/Ca is only rarely investigated in species producing high-Mg calcite (Evans et al., 2015; Van Dijk et al., 2017)". In the discussion (lines 356-357) we have replaced "has not been observed before" by "fits previously reported (differences in) partition coefficients for barium" and in the conclusions (now lines 484-485) we have replaced "are larger than previously thought" by "can be relatively large".

3. Section 2.2. Please state the approximate volume of seawater in which the cultures took place. As these cultures were performed in petri dishes, presumably the volume was relatively small? If so: (1) How was evaporation monitored and avoided? (2) It is likely that the foraminifera modified the carbonate chemistry of the seawater in between water exchanges (once per week). Was this monitored?

The Petri dishes contained approximately 10 ml of culture medium, which is now added to the Method section (line 131). This volume is obviously still very large in comparison to the elemental uptake of the foraminifera. Evaporation was minimized by replacing the medium every three days (and not once a week: line 153). The carbonate chemistry was not monitored within the Petri dishes and may in theory have been slowly changing during these three days. A simple calculation shows that calcification as such cannot have had a measurable effect on the total inorganic carbon concentration. The uptake

of DIC by the addition of a new chamber in juvenile foraminifera is in the order of 0.36 nmol (De Nooijer et al., 2009. Biogeosciences 6: 2669-2675). In 10 ml of our seawater (with approximately 2200 $\mu$mol DIC /l; Table 1) contained 22 $\mu$mol of DIC. Calcification of one new chamber therefore removes only $\sim$0.015% of the present DIC (0.36 nmol/22000 nmol). With 20 specimens per dish, only $\sim$0.3% of the DIC would have been removed if each of them produced one new chamber during those three days.

4. The phrasing of lines 148-149 implies that the cleaning procedure has an impact on measured Ba/Ca. Either rephrase or state what this impact is.

The impact of both cleaning with NaOCl and H2O2 is similar and relatively small (2-3 $\mu$mol/mol; Table 2 in Pak et al., 2004) compared to rinsing with de-ionized water. We have clarified this in the revised version of our manuscript (now lines 161-162).

5. There is far more detail given for the laser-ablation performed at the royal NIOZ. Whilst a reference is provided for the system at Utrecht University, it may be useful to state which LA and ICPMS systems were used and the wavelength of the laser for easy comparison. What is the accuracy and precision of the system used at Utrecht University?

As also noted by reviewer #1, we have extended the description of the platform used at the Utrecht University (lines 167-170). Relative standard deviation based on ablation of standard calcite material was 5% for Mg/Ca, as well as for Sr/Ca (lines 180-181).

6. Lines 164-169. Here the authors state that the final chamber of A. lessonii and the final two chambers of H. depressa did not yield reliable data because the walls are thin. However, Fig. 3 shows data for F and F-1 for both species. Is this a mistake? If not, these data should not be shown if they are not reliable.

We agree with the reviewer that this may be seen as contradictory. We have hance extended the text on this part clarifying 'unreliability' of measuring the F- and F-1 chambers. In short, shorter ablation profiles result in a higher 'within-chamber wall' Ba/Ca (as well as Mg/Ca, Sr/Ca, etc) variability. Therefore, the precision of the determined Ba/Ca is lower in thinner (i.e. built later in life) chambers. This is in line with a higher variability being observed between the F-chamber's average Ba/Ca (compare the SDs between F and F-5 in figure 3). To minimize the variability due to shorter ablation profiles, F- and F-1 chambers were omitted from the larger dataset. Unless they were extremely short (< 5 sec; which was the case in 4 out of 10 specimens for both Amphistegina and Heterostegina), they were analyzed for the dataset focusing on Ba/Ca as a function of chamber position (figure 3). One of the reasons is that including them is the only way to show, for example, that the variability between average Ba/Ca for F-chambers is higher than for (e.g.) F-2 chambers. The text in the method section is changed to account for the above considerations (now lines 181-184).

7. Lines 175-176. What is the possible source of Al and Mn contamination in cultured foraminifera?

We are unfortunately not entirely sure on this, although this is commonly observed in such studies by different laboratories working on similar studies. It may be an artifact of the analytical approach in which the first few laser pulses cause the material deposited in the ablation chamber to whirl up due to the plasma plume explosion. Alternatively, it may be a remnant from the foraminiferal cell material that accumulated these metals during the culturing. The pseudopodial network is known to cover the complete outer surface, for example during chamber formation and may leave traces at the shell surface after termination of the experiment. Since this is pure speculation, and monitoring for high Al and/or Mn counts is relatively standard procedure in laser-ablating fossil as well as recent foraminifera, we suggest to leave the text as it is on this point.

8. Section 2.4. Consider changing the phrase 'field specimens'.

We have changed this to 'aquarium samples'.

9. Lines 230-231. I agree that it is reasonable to assume no barium incorporation

when there is no barium in seawater. However, forcing a linear regression through the origin also assumes that seawater and shell Ba/Ca must be linearly related across the full range of seawater Ba/Ca ratios, which may not be the case. Consider that the H. depressa zoo aquarium sample may be in agreement with the cultures if the regression is not forced through the origin.

When including the aquarium samples in the linear regression analysis, for both species, the R2 decreases and the intercept with the y-axis increases. When including the aquarium samples in the linear regression and forcing the regression through (0, 0), the slopes hardly change (from 0.78 to 0.77 for H. depressa and from 0.33 to 0.32 for A. lessonii). This implies that the aquarium samples agree well with the experimental specimens when forcing the regression through the origin. Following the reviewer's suggestion, we have added this outcome to the manuscript (lines 265-270). As stated previously, the conditions under which the aquarium samples have formed their shells may have been different and since those conditions are less-well constrained than in our culturing experiment, we suggest to exclude them from the regression shown in figure 2.

10. Lines 232-233. Technically, if the regressions are not forced through the origin then there is not a single partition coefficient value, I suggest this is rephrased in terms of the seawater-shell Ba/Ca slopes.

We agree with the reviewer and have changed the text regarding the change in Ba/Cacc as a function of Ba/Casw when the linear regression is not forced through zero (lines 254-255).

11. Lines 241-242. Why is the aquarium seawater Ba/Ca higher than most natural seawater? If this is known it would be useful to state the reason. What is the meaning of the sentence starting on line 243? Is it an analytical problem or is there reason to suspect the aquarium seawater Ba/Ca ratio was not constant?

The Ba/Ca of the aquarium's seawater is indeed higher than that of the open ocean

(which is approximately 0.15 $\mu$mol/L Ba/ 10 mmol/L Ca, or 15 $\mu$mol Ba/mol Ca). In coastal waters, however, this ratio can easily be 2-3 times higher (see e.g. Shaw et al., 1998. GCA 62: 3047-3054), similar to the Ba/Ca of the zoo's water. 'Conditions' in this sentence refers to any chemical/ physical parameter that was not (accurately) measured over the life-time of the foraminifera that were used for Ba/Ca analysis. Many parameters (e.g. salinity, temperature) are regularly determined in the aquarium, but not as precise/ accurate as for our controlled growth experiment. Moreover, conditions (e.g. carbonate chemistry) within the coral rubble at the aquarium's floor (where the foraminifera were collected from) may differ from the water itself, where samples for chemical and physical monitoring were taken. To avoid the suggestion that 'conditions' in this sentence is interpreted as 'Ba/Casw', we have changed this sentence accordingly (lines 265-268).

12. Section 3.3. It would aid the interpretation of these interesting data if the reader had an idea of how much of the foraminifera the final five chambers represent. Approximately how many chambers were precipitated in culture? Consider adding a representative image.

For both species, the final ∼6 chambers represent approximately half of the outer whorl. Since older (i.e. smaller) chambers are completely covered by the newly added chambers of the outer whorl, it is difficult to estimate the percentage of the final ∼6 chambers of the total number of chambers. Including a picture of (ablated) specimens wouldn't allow such an estimate either.

13. Lines 259-260. The phrasing is confusing here. Ba/Ca increases in the first sentence but decreases in the second sentence. Rephrase for consistency.

The 'chamber position', 'chamber addition' and 'chamber number' and the directions in which these are represented, may have resulted in an apparent inconsistency here. We have rephrased the first sentence to make sure that Ba/Ca increases with subsequent new chambers added (Figure 3). The second sentence now reads: "...average

increase of 1.43 $\mu$mol/mol Ba/Cacc with every chamber added (Table 4)."

14. Lines 273-276. If the two slopes are significantly different, why combine the data from both species?

We agree with the reviewer and have deleted the last sentence of this section.

15. Section 4.2 and Figure 4. I am surprised that the range in Mg/Ca is so large, both within and between experiments, and this requires further explanation. For example, compare these H. depressa data to those reported in Raitzsch et al. [2010]. In that paper the Mg/Ca 2SD was 17 to 24 mmol mol-1 (âĹij10-20%), which is comparable to other studies reporting laser-ablation data. Here, some experiments are in line with this while others have a far larger range, for example H. depressa treatment D (âĹij110-190 mmol mol-1). The A. lessonii data are even more surprising, treatment D has a range from 30-140 mmol mol-1, and treatment C has a range of 30-120 mmol mol-1. Why is there so much variation compared to other studies? Could something in the experimental design have resulted in this? Is it a result of using juvenile foraminifera?

The total range in Mg/Ca reported here and the standard deviation in Mg/Ca reported previously (by e.g. Raitzsch et al.) are not directly comparable. In our dataset, the Mg/Ca in Heterostegina (average 152 mmol/mol) varies between ∼110 and 190, although the SD is less than 25 mmol, or ∼16% of the average Mg/Ca. A large difference between the minimum and maximum value does not necessarily result in a high relative standard deviation, particularly when datasets are relatively large (n= 133 for H. depressa). Therefore, the variability in Mg/Ca observed is not unusually high in this species and actually comparable to those reported earlier (e.g. Sadekov et al., 2008; De Nooijer et al., 2014a). For Amphistegina (average Mg/Ca = 64 mmol/mol, RSD = 47%, n=188), the variability is indeed relatively high, but as discussed in sections 4.4 and 4.5, the increase in Ba/Casw may have an effect on the incorporation of other elements (e.g. Mg) and therefore explain part of the variability in Mg/Ca. Within treatments (A-E), for example, relative variability in Mg/Ca is much lower (27, 28, 28, 36

and 37%). This has been added to the text (lines 293-294).

16. Line 363-366. Alternatively, van Dijk et al. [2017] showed that pCO2 does impact Ba incorporation in Amphistegina, so perhaps the microenvironment carbonate chemistry can help to explain these data.

We agree with the reviewer and have therefore added a sentence here (lines 367-368) with a reference to Van Dijk et al. (2017).

17. Line 428-431. This argument is not valid, we would not expect a doubling in O. universa Mg/Ca to exert a resolvable impact on Ba or Sr incorporation. For example, the DSr-Mg/Ca slope for inorganic calcite is $9.1 \times 10$-4 [Mucci & Morse, 1983], so that a change in shell Mg/Ca of 10 mmol mol-1 would result in a Sr/Ca increase of just âĹij0.1 mmol mol-1. If the relationship between DBa and Mg/Ca is similar, we would not observe this effect in Orbulina. It is visible in high-Mg species only because the shell Mg/Ca ratios are 1-2 order of magnitude higher than low-Mg foraminifera.

We agree and have deleted this sentence from the revised version of our manuscript.

18. Line 446. I think repeating the assertion that miliolids calcify intracellularly should be avoided. As stated a few lines later, it is intracellular only in the sense that calcification takes place from endocytosed seawater, which may well be the case for rotaliid foraminifera as well.

We agree that the distinction between biomineralization in rotaliids and milliolids is more subtle than stated previously. However, the contribution of ions (mainly DIC and Ca2+) delivered to the calcification space through cell membranes (see e.g. Nehrke et al., 2013; Toyofuku et al., 2017) suggests that the contribution of (unmodified) seawater differs greatly between these groups of foraminifera. We have altered the text in these sentences to reflect this more accurately (lines 478-481).

19. Line 449-450. van Dijk et al. [2017] report Na, Zn and Ba data for miliolid foraminifera.

We agree and have changed this into: "although so far mainly Sr/Ca and Mg/Ca of Miliolid foraminifera have been published".

20. Line 458. You could also reference Hoffmann et al. [2014] Geology 42:579 and Evans et al. [2015] G-cubed 16:2598.

We have added these references.

21. Figure 3. It would be more intuitive to plot the final chamber on the right hand side of the graphs, so that time goes forward from left to right.

We have reversed the order of the chamber number in this figure.

22. Figure 4. Please use symbols for the two species that are easier to distinguish. Consider plotting the slopes discussed in the text.

We have changed the symbols for H. depressa to increase the contrast with those of A. lessonii. In addition, we added the regression lines and their formulas to the figure too.

Typos 1. Line 183. 'naturel'. 2. Line 201. 'costume-built'. 3. Line 221. Write out '2'. 4. Line 274. Delete 'in a'. 5. Line 359. 'maybe'.

All typos were corrected in the new version of our manuscript.
* * *